# DATA-FREE PRUNING OF SELF-ATTENTION LAYERS IN LLMS

## ABSTRACT

Many self-attention sublayers in large language models (LLMs) can be removed with little to no loss. We attribute this to the *Attention Suppression Hypothesis*: during pre-training, some deep attention layers learn to mute their own contribution, leaving the residual stream and the MLP to carry the representation. We propose **Gate-Norm**, a one-shot, weight-only criterion that ranks attention sub-layers by query–key coupling and removes the least coupled ones—*requiring no calibration data, no forward passes, no fine-tuning, and no specialized kernels*. On 40-layer, 13B-parameter LLAMA models, Gate-Norm prunes the model under a second. Pruning 8–16 attention sublayers yields up to $1.30\times$ higher inference throughput while keeping average zero-shot accuracy within $2\%$ of the unpruned baseline across BoolQ, RTE, HellaSwag, WinoGrande, ARC-Easy/Challenge, and Open-BookQA. Across these settings, Gate-Norm matches data-driven pruning methods in accuracy while being $\sim 1000\times$ faster to score layers, enabling practical, data-free compression of LLMs.

## 1 INTRODUCTION

Large language models (LLMs) have grown from millions to **hundreds of billions** of parameters, providing success in translation, code generation, and reasoning (Brown et al., 2020; Meta-AI, 2023a; OpenAI, 2023). However, this growth increases inference latency, energy consumption, and carbon emissions, driving up deployment costs and environmental impact (Luccioni et al., 2023). To alleviate these burdens, prior work has compressed models along two axes: *precision*, via quantizing weights to 8, 4, or 2 bits (Frantar et al., 2022; Lin et al., 2024), and *width*, via weight pruning to remove redundant parameters (Frantar & Alistarh, 2023; Sun et al., 2024; Yin et al., 2023). These approaches, however, deliver significant speed-ups only on hardware with specialized low-precision or sparse-matrix kernels and often require fine-tuning to restore accuracy (Eccles et al., 2025). Meanwhile, the third axis—*depth*, i.e. the number of layers remains largely under-explored for LLMs.

An LLM first maps input tokens to embeddings, then processes them through $N$ Transformer blocks (Vaswani et al., 2017). Each block consists of (i) a multi-head self-attention sublayer that performs *token mixing* - each token attends to every other token, and (ii) an *MLP* (a two-layer feed-forward network) that performs *channel mixing* independently at each token. In LLAMA-7B (Meta-AI, 2023b), for example, an attention layer contains about 67 million parameters, while its MLP layer holds roughly 135 million. Despite having half the parameter count of an MLP layer, attention still dominates runtime - taking nearly twice as long as the MLP at $n \approx 3000$ and about three times longer by $n \approx 7000$, due to its $O(n^2)$ dependence on sequence length $n$ (Figure 1). Attention is clearly the runtime bottleneck. Consequently, pruning even a few attention layers, or dropping entire Transformer blocks, translates directly into substantial inference speed-ups on any hardware.

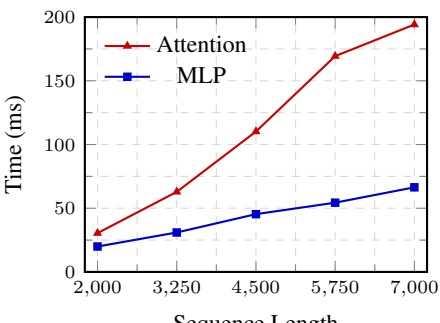

Figure 1: Inference time of different layer types vs sequence length for LLAMA-7B.

Recent work has targeted *structural* redundancy (Gromov et al., 2024) instead of weight pruning. SHORTGPT (Men et al., 2024) computes cosine similarity between each block's input and output on a calibration dataset: high similarity implies the block changes its input minimally and can be removed. However, accuracy drops significantly after pruning more than four blocks. The same similarity test is applied at sublayer granularity (attention vs. MLP), showing that many attention layers are more redundant than MLP layers (He et al., 2024). Crucially, both works mentioned above depend not only on **thousands of calibration tokens** and **multiple forward-passes**, but also on **grid-searching across diverse datasets** (e.g. C4 (Raffel et al., 2020), CodeAlpaca[1], MathInstruct (Xiang Yue, 2023), and LIMA (Zhou et al., 2023)) to select the corpus that best preserves downstream performance, increasing the compute overhead. In contrast,

---

[1] https://huggingface.co/datasets/sahil2801/CodeAlpaca-20k

an ideal *data-free* criterion - one that inspects only the model's learned weights - enables *instantaneous, on-device pruning* without any external data, hyperparameter tuning, or dataset leakage.

*Figure 2* highlights the structural outcome of removing a redundant attention sublayer: the residual stream bypasses self-attention and feeds the MLP directly.

We ask a deeper question: *why* do many attention layers contribute so little? We propose the **Attention Suppression Hypothesis**: during pre-training, deep attention sublayers learn to *mute* their outputs, producing near-zero updates so the residual connection carries the representation unchanged into the MLP. Crucially, a suppressed attention layer leaves a distinct pattern in its own weights, allowing us to detect redundancy *without any data*.

Exploiting this insight, we introduce a *data-free* importance score, *Gate-Norm*, computed directly from attention weights, which achieves over $1,000\times$ speedup compared to data-driven scoring methods. This millisecond-scale overhead enables practical, on-the-fly pruning in large-scale deployments, whereas calibration-based methods would be prohibitively slow for on-demand compression. It is to be noted that even in accelerator-free environments, Gate-Norm runs entirely

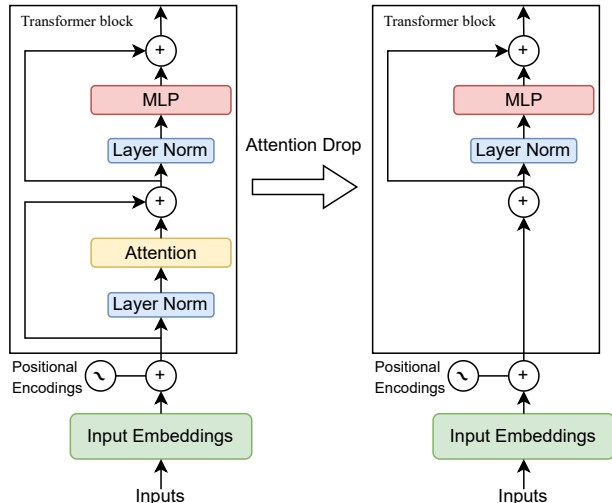

Figure 2: **Visualising attention drop.** Removing the self-attention branch leaves a direct residual path feeding the MLP.

within the limits of system RAM and no GPUs or forward passes are needed. A 13 B-parameter model can be pruned in **30 s** on a standard CPU. In contrast, data-driven pipelines require thousands of tokens and repeated forward-passes, rendering them prohibitively slow or infeasible without specialized hardware.

Removing one-third of attention sublayers preserves perplexity within **1%** of baseline, improves zero-shot accuracy on BoolQ, RTE, HellaSwag, WinoGrande, ARC-Easy/Challenge, and OpenBookQA, and increases end-to-end throughput by up to **1.4×**.

In summary, our key contributions are as follows:

- We formulate the *Attention Suppression Hypothesis*, explaining why deep self-attention sublayers become functionally inert.

- We propose the first fully *data-free*, millisecond-scale importance score for attention layers and a one-shot pruning algorithm.

- We validate our approach on large LLAMA checkpoints, achieving substantial latency reductions while preserving baseline perplexity and zero-shot performance.

## 2 RELATED WORK

### 2.1 PRELIMINARIES AND NOTATION

We consider a decoder-only Transformer of depth $L$, which maps an input token sequence to contextualized embeddings via $L$ identical blocks. At block index $\ell$, the representation is $X_\ell \in \mathbb{R}^{B \times S \times D}$ where $B$ is batch size, $S$ sequence length, and $D$ hidden dimension. Each block contains two residual layers:

**Self-attention layer:** This sublayer allows each token to 'attend' to every other token in the sequence, performing *token mixing* to aggregate contextual information across positions. It is parameterized by trainable projection matrices $W_{q,\ell}$, $W_{k,\ell}$, and $W_{v,\ell}$ for queries, keys, and values, as well as $W_{o,\ell}$ for the output projection.

$$Z_\ell = \text{LayerNorm}(X_\ell), \tag{1}$$

$$\text{AttnOut}_\ell = \text{MHA}(Z_\ell), \tag{2}$$

$$Y_\ell = X_\ell + \text{AttnOut}_\ell. \tag{3}$$

Here, multi-head attention (MHA) splits $D$ into $H$ heads of size $d = D/H$. For each head $h$, we compute:

$$Q_\ell^{(h)} = Z_\ell W_{q,\ell}^{(h)}, \qquad\qquad K_\ell^{(h)} = Z_\ell W_{k,\ell}^{(h)}, \qquad\qquad (5)$$

$$V_\ell^{(h)} = Z_\ell W_{v,\ell}^{(h)}, \qquad\qquad L_\ell^{(h)} = \frac{Q_\ell^{(h)}(K_\ell^{(h)})^\top}{\sqrt{d}}, \qquad\qquad (6)$$

$$\Pi_\ell^{(h)} = \mathrm{softmax}\big(L_\ell^{(h)}\big), \quad \mathrm{AttnOut}_\ell^{(h)} = \Pi_\ell^{(h)} V_\ell^{(h)} W_{o,\ell}^{(h)}. \qquad\qquad (7)$$

Concatenating across heads yields

$$\mathrm{AttnOut}_\ell \in \mathbb{R}^{B \times S \times D}.$$

**Feed-forward (MLP) layer:** Following attention, each token's representation is independently transformed by a two-layer MLP, mixing features across channels but not across positions. This layer is parameterized by $W_{1,\ell}$ and $W_{2,\ell}$.

$$U_\ell = \mathrm{LayerNorm}(Y_\ell), \qquad\qquad (4)$$

$$\mathrm{MLP}_\ell = W_{2,\ell} \, \mathrm{GELU}\big(W_{1,\ell} \, U_\ell\big), \qquad\qquad (5)$$

$$X_{\ell+1} = Y_\ell + \mathrm{MLP}_\ell. \qquad\qquad (6)$$

Here, the MLP expands the token embedding to a higher-dimensional hidden layer, applies a nonlinearity, and projects back to dimension $D$. This enables per-token feature mixing and further transformation before passing to the next block.

### 2.2 COSINE-SIMILARITY IMPORTANCE

To identify redundant blocks or layers, prior work measures how little they change their inputs via cosine similarity (Men et al., 2024; Gromov et al., 2024). For block-level importance, let $X_{\ell+1,b,t}$ be the output of block $\ell$ for batch index $b$ and token $t$. The per-token similarity is

$$\cos\big(X_{\ell,b,t}, X_{\ell+1,b,t}\big) = \frac{X_{\ell,b,t}^\top X_{\ell+1,b,t}}{\|X_{\ell,b,t}\| \, \|X_{\ell+1,b,t}\|}, \qquad\qquad (7)$$

which equals 1 when the block leaves its input unchanged. Averaging over all non-padding tokens ($T$ total) results in

$$\bar{c}_\ell = \frac{1}{T} \sum_{b,t} \cos\big(X_{\ell,b,t}, X_{\ell+1,b,t}\big), \qquad\qquad \mathrm{Imp}_\ell^{\mathrm{block}} = 1 - \bar{c}_\ell. \qquad\qquad (8)$$

Blocks with low $\mathrm{Imp}_\ell^{\mathrm{block}}$ (high similarity) have limited representational impact and can be dropped.

**ShortGPT:** The ShortGPT framework (Men et al., 2024) uses the importance of block level $\mathrm{Imp}_\ell^{\mathrm{block}}$ to rank and remove 4 of 40 blocks in Llama 13B while offering comparable accuracy of downstream tasks.

### 2.3 LAYER-LEVEL ATTENTION VS. MLP PRUNING

ShortGPT treats each block as an atomic unit, but redundancy can be finer-grained.

**Not All Attention Is Needed:** He *et al.* (He et al., 2024) apply the cosine-similarity measure inside each block at sublayer granularity. Defining

$$Y_{\ell,b,t}^{\mathrm{attn}} = X_{\ell,b,t} + \mathrm{AttnOut}_{\ell,b,t}, \qquad\qquad (9)$$

$$Y_{\ell,b,t}^{\mathrm{mlp}} = Y_{\ell,b,t}^{\mathrm{attn}} + \mathrm{MLP}_{\ell,b,t}, \qquad\qquad (10)$$

they compute

$$\mathrm{Imp}_\ell^{\mathrm{attn}} = 1 - \frac{1}{T} \sum_{b,t} \cos\big(X_{\ell,b,t}, Y_{\ell,b,t}^{\mathrm{attn}}\big), \qquad\qquad (11)$$

$$\mathrm{Imp}_\ell^{\mathrm{mlp}} = 1 - \frac{1}{T} \sum_{b,t} \cos\big(Y_{\ell,b,t}^{\mathrm{attn}}, Y_{\ell,b,t}^{\mathrm{mlp}}\big). \qquad\qquad (12)$$

They observe that, for many mid- and late-stage layers, $\mathrm{Imp}_\ell^{\mathrm{attn}} \ll \mathrm{Imp}_\ell^{\mathrm{mlp}}$, indicating that attention sublayers are far more redundant than their MLP counterparts. Given that attention is the primary runtime bottleneck, this finding motivates our choice to prune attention sublayers, delivering substantial compute savings with negligible impact on performance.

## 2.4 QUANTIZATION AND UNSTRUCTURED WEIGHT PRUNING

Simultaneous efforts target *parameter-level* redundancy:

- **Quantization** reduces weight and activation precision (e.g. INT8, INT4, INT2), lowering memory footprint and accelerating matrix-vector kernels on hardware that supports low-bit arithmetic (Frantar et al., 2022; Lin et al., 2024). However, most commodity GPUs provide only limited INT8 support, and further gains at INT4 or INT2 often require specialized accelerators or custom kernels (Eccles et al., 2025). Moreover, quantized models typically need calibration passes or light fine-tuning to recover accuracy lost from reduced precision.
- **Weight pruning** removes individual weights based on Hessian- or activation-based criteria (Han et al., 2015; Singh & Alistarh, 2020; Yu et al., 2022; Benbaki et al., 2023). Techniques, such as SparseGPT (Frantar & Alistarh, 2023), Wanda (Sun et al., 2024), and OWL (Yin et al., 2023), achieve over 50% sparsity with minimal accuracy degradation, but the resulting random sparsity patterns do not map efficiently to standard hardware. Achieving actual runtime speedups often relies on N:M block-sparsity formats (e.g. NVIDIA Ampere's 2:4 sparsity) or custom sparse kernels (Eccles et al., 2025), which have only become available on recent accelerators.

Parameter-level methods can effectively compress weights but depends on the availability of specialized hardware or sparsity formats to achieve inference speedups. In contrast, our work targets *depth redundancy*: by removing entire attention layers, we achieve substantial reductions in compute and memory without requiring low-bit support, sparse-kernel libraries, or any special accelerator features.

Several recent methods also modify the depth or structure of LLMs. LLM-Pruner (Ma et al., 2023) performs task-agnostic structural pruning by grouping coupled modules and ranking them with gradient-based importance computed on a calibration set, then recovers performance via LoRA fine-tuning. D-LLM (Jiang et al., 2024) and SkipGPT (Zhao et al., 2025) instead learn dynamic token-level routing policies that decide at inference time which layers (and, in SkipGPT, whether attention or MLP blocks) to execute, trained with supervision and parameter-efficient fine-tuning. AdaInfer (Fan et al., 2024) is an early-exit scheme that trains lightweight classifiers on intermediate statistics to predict an input-specific stopping layer without changing backbone weights. In contrast, our approach uses a *static*, purely weight-based criterion (GateNorm-Attn) to prune self-attention sublayers once, without calibration data or retraining in our main setting, and is therefore complementary to these dynamic, data-driven depth-adaptation schemes.

## 3 ATTENTION SUPPRESSION HYPOTHESIS

Prior pruning studies (Men et al., 2024; He et al., 2024) show that attention layers with the lowest cosine-similarity importance can be removed with minimal loss. However, these works do not explain *why* deep attention layers become redundant. We hypothesize that, during pretraining, later attention layers learn to *actively suppress* their own updates, producing nearly zero, so that all representational work is carried by the identity residual and the MLP. We now formalize this hypothesis and validate it empirically.

### 3.1 EMPIRICAL OBSERVATION OF ATTENTION SUPPRESSION

**Cosine-Similarity Importance Goes to Zero:** The attention-layer importance score is defined as (He et al., 2024)

$$\text{Imp}_\ell^{\text{attn}} = 1 - \frac{1}{T} \sum_{b,t} \cos\big(X_{\ell,b,t}, Y_{\ell,b,t}^{\text{attn}}\big), \tag{13}$$

where

$$Y_{\ell,b,t}^{\text{attn}} = X_{\ell,b,t} + \text{AttnOut}_{\ell,b,t},$$

and $\text{AttnOut}_{\ell,b,t}$ is the post-LayerNorm attention update. They observe

$$\text{Imp}_\ell^{\text{attn}} \longrightarrow 0 \quad \text{for} \quad \ell \geq \ell_0, \tag{14}$$

i.e. the cosine similarity $\cos(X_{\ell,b,t}, Y_{\ell,b,t}^{\text{attn}}) \to 1$ in deep layers.

**From Cosine to Zero Update:** A cosine similarity of 1 between a vector $X$ and its residual-augmented version $X + U$ implies $U$ must vanish. Indeed,

$$\cos(X, X + U) = \frac{X^\top (X + U)}{\|X\| \, \|X + U\|} = \frac{\|X\|^2 + X^\top U}{\|X\| \, \|X + U\|}. \tag{15}$$



Figure 3: Attention-to-input norm ratio $r_\ell$ across layers 1–40 in the 40-layer LLaMA-13B. Early layers exhibit high ratios, mid layers plateau around 0.3, and deeper layers collapse toward zero, confirming that later attention updates become negligible.

As $\cos(X, X + U) \to 1$, we require

$$\|X + U\| \to \|X\| \quad \text{and} \quad X^\top U \to 0.$$

By the reverse triangle inequality,

$$\big| \|X + U\| - \|X\| \big| \leq \|U\|, \tag{16}$$

so $\|X + U\| \to \|X\|$ forces $\|U\| \to 0$. Hence,

$$\|\text{AttnOut}_\ell\| \to 0 \quad \text{for} \quad \ell \geq \ell_0, \tag{17}$$

demonstrating that deep attention layers have learned during pretraining to suppress their own updates and forward inputs unchanged.

### 3.2 ATTENTION-TO-INPUT NORM RATIO

**Definition:** We quantify the relative strength of the self-attention update versus the residual path by defining the *attention-to-input norm ratio*

$$r_\ell = \frac{\sum_{b,t} \|\text{AttnOut}_{\ell,b,t}\|}{\sum_{b,t} \|X_{\ell,b,t}\|}, \tag{18}$$

where $X_{\ell,b,t}$ is the pre-LayerNorm input at layer $\ell$ for batch index $b$ and token position $t$, and $\text{AttnOut}_{\ell,b,t}$ is the corresponding post-LayerNorm attention update. Since the MLP block receives the sum $X_{\ell,b,t} + \text{AttnOut}_{\ell,b,t}$, $r_\ell$ directly measures the fraction of the MLP's input norm contributed by self-attention. In particular, $r_\ell = 0 \iff$ the MLP sees only the residual path, while larger $r_\ell$ indicates greater attention influence.

**Empirical Validation:** We compute $r_\ell$ on a calibration set of 1,024 token sequences for the 40-layer LLAMA-13B model. As plotted in Figure 3, $r_\ell$ has a steep drop initially followed by a steady drop with depth: - $r_1 \approx 1.4$: the first layer's attention update norm exceeds the residual, so attention dominates the MLP input. - $r_{2-4} \approx 0.4$–$0.5$: attention still contributes substantially but with diminishing strength. - $r_{5-16} \approx 0.3$: a stable mid-network plateau of moderate contribution. - $r_{\ell > 22} < 0.15$, reaching $\approx 0.08$ at layer 40: attention contributes less than 10 % of the input norm, so over 90% flows through the residual. This monotonic decay of $r_\ell$ confirms that, beyond layer 22, self-attention sublayers produce negligible updates, validating the Attention Suppression Hypothesis and motivating our data-free pruning criterion. In the next section, we show how to detect this suppression using only a layer's weights.

Taken together, the decay of attention-layer cosine importance with depth and the collapse of the attention-to-input norm ratio in late layers provide direct empirical support for the Attention Suppression Hypothesis, independently of any downstream pruning experiments.

## 4 DATA-FREE PRUNING OF ATTENTION LAYERS USING GATE-NORM PROXY

The analysis in Section 3 highlighted that, beyond a certain depth, many self-attention sublayers in LLMs effectively "mute" their outputs, forwarding inputs unchanged through the residual path into the MLP. Rather than identifying these dormant layers via a data dependent way via forward-passes over thousands of tokens, we propose a fundamentally *data-free* approach that inspects only the pre-trained query and key weights.

Our proposed method, *Gate-Norm*, computes for each self-attention sublayer a single proxy score—based on the $W_q$ and $W_k$ matrices. This weight-only heuristic correlates tightly with the degree of attention suppression, enabling a one-shot ranking and pruning of layers purely by their learned parameters. Gate-Norm executes in the order of milliseconds on modern GPUs, enabling dynamic, on-the-fly compression viable for efficient inference. Crucially, it also runs seamlessly in accelerator-free environments, completing the same pruning routine in under 30 seconds on a CPU using only system RAM, whereas calibration-based, data-driven pipelines require thousands of tokens, repeated forward-passes, and substantial GPU memory, rendering them impractical without specialized hardware setups. By eliminating both data dependence and inference overhead, Gate-Norm is the first fully data-independent, forward-pass-free algorithm for pruning attention layers, uniquely suited for on-device and privacy-constrained deployments.

**Key Hypothesis:** *If a layer's query–key coupling is weak, it cannot effectively mix information across tokens, so its attention output vanishes ($\|\mathrm{AttnOut}_\ell\| \to 0$), implying $Y_\ell = X_\ell + \mathrm{AttnOut}_\ell \approx X_\ell$.*

In other words, self-attention only modifies embeddings via the $QK^\top$ interaction; if that interaction is small, the layer effectively acts as the identity.

We formalize this in three steps (complete derivations in Appendix A):

**1. Gate-Norm Proxy:** For the $\ell^{\mathrm{th}}$ attention layer, collect all heads' query and key weight matrices:

$$W_{q,\ell},\, W_{k,\ell} \,\in\, \mathbb{R}^{D \times D}. \tag{19}$$

Define the *gate matrix*

$$M_\ell = W_{q,\ell}\, W_{k,\ell}^\top, \tag{20}$$

and its Frobenius norm—the *gate-norm*:

$$m_\ell = \|M_\ell\|_F. \tag{21}$$

Intuitively, $m_\ell$ measures the maximum bilinear "mixing strength" any two normalized token vectors can induce in the raw attention logits.

**2. From Gate-Norm to Uniform Attention:** After LayerNorm, let token embeddings be $z_i, z_j \in \mathbb{R}^D$. The raw attention logits satisfy

$$L_{ij} = \frac{1}{\sqrt{d_h}}\, z_i^\top M_\ell\, z_j, \quad |L_{ij}| \le C_0\, m_\ell, \tag{22}$$

where $d_h = D/H$ and $C_0 = O(1)$. If $m_\ell \ll 1$, a first-order expansion of the softmax shows each row is (nearly) uniform:

$$A_{ij} = \frac{e^{L_{ij}}}{\sum_k e^{L_{ik}}} = \frac{1}{S} + O(m_\ell), \tag{23}$$

where $S$ is the sequence length. Thus deviations from exact uniform attention ($\frac{1}{S}$) are $O(m_\ell)$.

**3. From Uniformity to Vanishing Cosine-Importance:** Let $V_j W_{o,\ell}$ denote the values plus output projection. The per-token update then decomposes as

$$\mathrm{AttnOut}_{\ell,i} = \sum_{j=1}^{S} A_{ij}\,(V_j W_{o,\ell}) = u + \Delta_i, \quad u = \frac{1}{S}\sum_{j=1}^{S} V_j W_{o,\ell}, \quad \|\Delta_i\| = O(m_\ell). \tag{24}$$

Subtracting the shared shift $u$ (by mean-centering both $X_\ell$ and $Y_\ell$) leaves each centered output differing only by $\Delta_i$. We then invoke the "cosine-law" inequality

$$1 - \cos(x,\, x+v) \;\le\; \frac{\|v\|}{\|x\|}, \tag{25}$$

which holds for any nonzero $x$ and perturbation $v$. Applying this with $x = \tilde{X}_{\ell,i}$ and $v = \Delta_i$, using $\|\tilde{X}_{\ell,i}\| = \Theta(1)$, yields a per-token cosine deviation of $O(m_\ell)$. Averaging over all $T$ tokens then gives

$$\mathrm{Imp}_\ell^{\mathrm{attn}} = 1 - \cos(X_\ell, Y_\ell) = O(m_\ell). \tag{26}$$

**One-Shot Gate-Norm Pruning:** To prune, compute $\{m_\ell\}_{\ell=1}^{L}$ using Equation 21, sort layers by increasing $m_\ell$, and disable attention updates in the smallest $N$ layers (where $Y_\ell \approx X_\ell$) as outlined in Algorithm 1. This single-pass procedure runs in a few milliseconds on billion-parameter models, requires no data, and matches or outperforms data-driven baselines (Section 5).

---

**Algorithm 1** Procedure for One-Shot Gate-Norm Pruning

---

**Require:** Query/key weights $\{W_{q,\ell}, W_{k,\ell}\}_{\ell=1}^{L}$, prune count $N$
1: **for** $\ell = 1$ **to** $L$ **do**
2:      $m_\ell \leftarrow \left\| W_{q,\ell} W_{k,\ell}^\top \right\|_F$                                               ▷ Equation 21
3: **end for**
4: $\pi \leftarrow \text{argsort}(m_1, \ldots, m_L)$                                                           ▷ ascending
5: **for** $i = 1$ **to** $N$ **do**
6:      Disable attention update at layer $\pi[i]$                ▷ i.e. skip or set $\text{AttnOut}_{\pi[i]} = 0$
7: **end for**
8: **return** Pruned model

---

## 5 EXPERIMENTS

We evaluate Gate-Norm on two LLaMA-13B variants (v1 and v2) (Meta-AI, 2023a;b), comparing against: (1) *Short-GPT block removal* (Men et al., 2024), which drops entire Transformer blocks based on similarity scores; (2) *Data-driven attention pruning* (He et al., 2024), which removes individual attention sublayers via similarity scores; (3) *Random block removal*, which randomly selects and drops $N$ full Transformer blocks; and (4) *Random attention removal*, which randomly selects and disables $N$ attention sublayers. All reported numbers are averaged over 3 runs.

We measure: **Perplexity** on WikiText-2 as layers are removed (Section 5.1), **Zero-shot accuracy** on standard benchmarks (BoolQ, RTE, HellaSwag, WinoGrande, ARC-Easy/Challenge, OpenBookQA; Section 5.2), and **Pruning latency**, i.e. time to compute importance scores and disable layers (Section 5.3).

**Hardware:** NVIDIA RTX A6000 GPU (48 GB, 10 752 CUDA cores, 336 Tensor cores, 309 TFLOPS peak) paired with an AMD EPYC 7713P 64-core CPU. **Software:** Ubuntu 20.04.6 LTS, Python 3.8.10, PyTorch 2.1.0.

### 5.1 PERPLEXITY EVALUATION

Perplexity is evaluated on the WikiText-2 (Merity et al., 2016) as attention layers are progressively removed. For both LLaMA-13B v1 and v2, we drop 1, 4, 8, 12, 16, and 20 attention layers under four strategies: Data-driven attention pruning, Gate-Norm attention removal (ours), random attention removal, and ShortGPT block removal. Figure 4 plots WikiText-2 perplexity versus number of dropped layers for v1 (left column) and v2 (right column), with full-range curves in the top row and zoomed-in comparisons in the bottom row.

In LLaMA-13B v1, the baseline WikiText-2 perplexity is about 5.10. Removing 4–7 attention layers raises perplexity only slightly under data-driven pruning (5.25–5.46), and Gate-Norm closely tracks or slightly improves these values (e.g., at 7 layers: 5.37 vs. 5.46). With 10–16 layers removed, perplexity grows gradually (data-driven from 6.3 to 8.6; Gate-Norm from 5.60 to 6.93), with Gate-Norm staying near or below the calibration curve. Beyond 16 layers, perplexity rises sharply for both attention-based methods; random attention removal is catastrophic (perplexity in the hundreds to thousands), while block removal is intermediate—already worse than attention pruning at moderate budgets and degrading faster at larger budgets.

LLaMA-13B v2 shows the same pattern: the baseline is about 4.92 and increases modestly under moderate pruning (data-driven 5.14–5.28), with Gate-Norm closely following or marginally outperforming; random attention removal again causes large jumps in perplexity, and block removal lags attention-based methods across budgets.

**Key Observations (Perplexity)**: On WikiText-2, Gate-Norm matches data-driven pruning across both LLaMA-13B variants at practical budgets (about 4–16 layers) and clearly outperforms random and block removals; beyond roughly 16 layers, all methods degrade rapidly.

**Additional models.** To assess whether attention suppression and Gate-Norm-based pruning extend beyond the original LLaMA-13B checkpoints, we further evaluate Vicuna-7B Chiang et al. (2023), Vicuna-13B Chiang et al. (2023), and LLaMA-3.1-8B Meta-AI (2024) on WikiText-2 under the same one-shot, data-free protocol. For each model we progressively remove 1, 4, 7, 10, and 13 attention sublayers using three strategies: random attention removal, a data-driven cosine-similarity baseline, and our data-free Gate-Norm-Attn criterion. As summarised in Table 1, random pruning adversely impacts perplexity once pruning becomes even moderately aggressive, whereas both principled schemes maintain low perplexity across a wide range of budgets. More importantly, Gate-Norm consistently matches

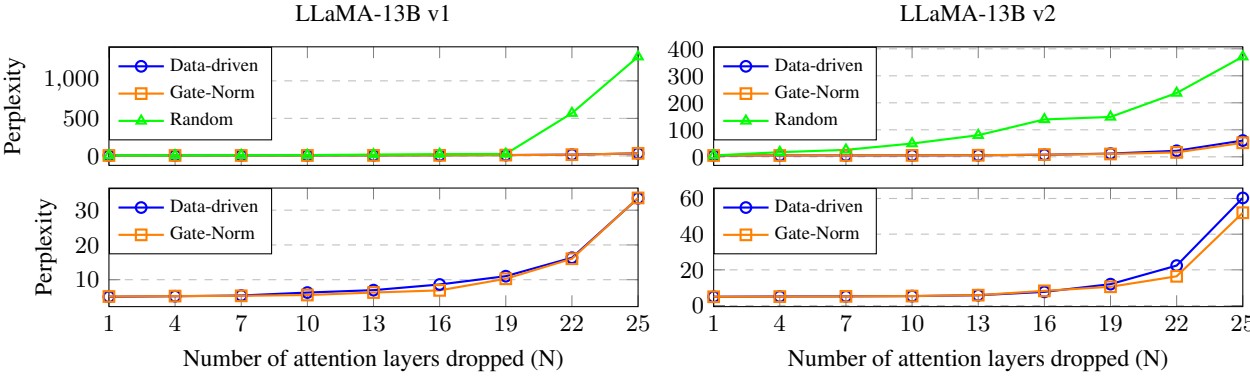

Figure 4: WikiText-2 Perplexity vs. number of dropped Attention layers on LLaMA-13B v1 (left) and v2 (right), up to 25 layers. Top row: full-range curves (including Random). Bottom row: zoomed-in comparison (Data-driven vs Gate-Norm).

Table 1: WikiText-2 perplexity on additional checkpoints under attention pruning. Lower is better. "Random" denotes random attention-layer removal; "Data-driven" is a cosine-similarity baseline; "Gate-Norm" is our data-free score.

| Method | Vicuna-7B | | | | | Vicuna-13B | | | | | LLaMA-3.1-8B | | | | |
|---|---|---|---|---|---|---|---|---|---|---|---|---|---|---|---|
| | # pruned attention layers | | | | | # pruned attention layers | | | | | # pruned attention layers | | | | |
| | 1 | 4 | 7 | 10 | 13 | 1 | 4 | 7 | 10 | 13 | 1 | 4 | 7 | 10 | 13 |
| Random | 6.82 | 7.84 | 13.84 | 355.87 | 473.40 | 6.00 | 10.63 | 67.09 | 587.90 | 594.83 | 6.51 | 8.11 | 14.59 | 54.51 | 98.55 |
| Data-driven | 7.23 | 7.62 | 8.78 | 12.55 | 20.46 | 5.99 | 6.11 | 6.24 | 6.39 | 7.54 | 6.35 | 6.83 | 7.39 | 9.79 | 16.48 |
| Gate-Norm | 7.23 | 7.42 | 8.57 | 10.02 | 16.64 | 5.97 | 6.05 | 6.24 | 6.47 | 7.05 | 6.30 | 6.67 | 7.74 | 11.55 | 15.65 |

or slightly outperforms the data-driven attention baseline on all three checkpoints, reinforcing that a fully data-free, weight-only score can recover the same high-confidence pruning decisions across a diverse range of models.

**Which layers are removed?** Figure 5 shows 16 attention sub-layers removed in each model–method pair. Both pruning strategies concentrate on pruning the later layers of the 40-layer stack, eliminating 12 identical layers in v1 and 14 in v2; the similarity in pruning decisions explains the near-overlap of the perplexity curves in Figure 4. Gate-Norm, driven solely by weight norms, initiates pruning earlier—around layer 20 in v1 and layer 23 in v2—and therefore retains a few deeper layers (e.g., layer 37 in v2), whereas the calibration-based rule removes a more contiguous band near the top (layers 25–40 in v1 and 24–39 in v2). The close alignment of these selections indicates that a simple weight-only heuristic can replicate almost all high-confidence pruning decisions without the forward-pass calibration.

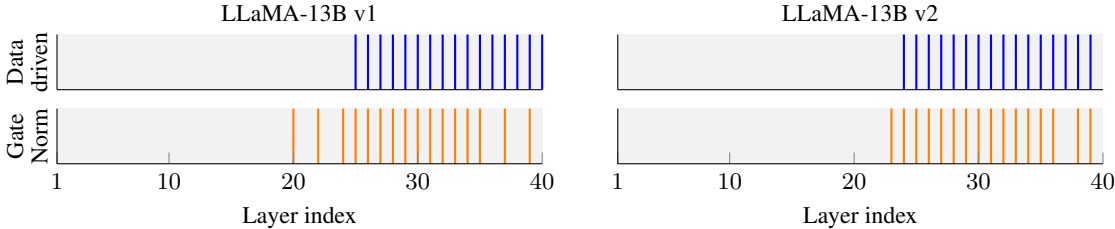

Figure 5: Pruned attention sub-layers in the 40-layer LLaMA-13B models. Columns compare model variants (v1 (left) vs. v2 (right)); rows compare pruning strategies (data-driven, top; data-free / Gate-Norm, bottom). Vertical bars indicate the attention sub-layers removed by each strategy.

## 5.2 ZERO-SHOT ACCURACY EVALUATION

Zero-shot performance on seven NLP benchmarks (BoolQ, RTE, HellaSwag, WinoGrande, OpenBookQA, ARC-Easy, ARC-Challenge) is summarized in Table 2 for LLaMA-13B v1 and v2. The unpruned baselines average about

| **LLaMA-13B v1** | | | | | | | | | | | |
|---|---|---|---|---|---|---|---|---|---|---|---|
| Method | #Pruned | Type | BoolQ | RTE | HellaSwag | WinoG | ARC-E | ARC-C | OpenBookQA | Avg Acc | SpeedUp | Data Required |
| Baseline | 0 | – | 77.86 | 70.40 | 78.10 | 73.01 | 69.28 | 47.18 | 43.80 | 65.66 | 1.00× | – |
| Random-Block | 4 | Block | 66.62 | 72.67 | 44.62 | 67.96 | 67.51 | 41.40 | 76.33 | 62.45 | 1.10 | ✗ |
| Random-Block | 8 | Block | 60.48 | 65.30 | 35.07 | 60.93 | 57.40 | 39.60 | 65.84 | 54.95 | 1.25 | ✗ |
| ShortGPT-Block | 4 | Block | 68.43 | 76.21 | 44.11 | 73.01 | 68.95 | 42.20 | 61.01 | 61.99 | 1.10 | ✓ |
| ShortGPT-Block | 8 | Block | 66.58 | 72.00 | 41.98 | 70.64 | 62.45 | 41.60 | 37.86 | 56.16 | 1.25 | ✓ |
| Random-Attn | 4 | Attention | 78.62 | 67.51 | 78.03 | 73.24 | 69.57 | 47.27 | 43.80 | 65.44 | 1.06 | ✗ |
| Random-Attn | 8 | Attention | 75.35 | 65.70 | 75.66 | 71.27 | 67.30 | 43.43 | 41.80 | 62.93 | 1.12 | ✗ |
| Random-Attn | 16 | Attention | 62.32 | 54.15 | 52.33 | 60.06 | 49.20 | 31.40 | 36.20 | 49.38 | 1.30 | ✗ |
| Data-driven-Attn | 4 | Attention | 77.89 | 70.04 | 78.03 | 73.09 | 69.57 | 47.61 | 44.00 | 65.75 | 1.06 | ✓ |
| Data-driven-Attn | 8 | Attention | 77.71 | 63.18 | 77.72 | 72.45 | 68.94 | 46.50 | 43.60 | 64.30 | 1.12 | ✓ |
| Data-driven-Attn | 16 | Attention | 78.10 | 68.59 | 76.73 | 72.69 | 64.65 | 44.20 | 43.20 | 64.02 | 1.30 | ✓ |
| Gate-Norm-Attn (Ours) | 4 | Attention | 78.01 | 65.34 | 78.20 | 72.61 | 69.65 | 46.93 | 44.20 | 64.99 | 1.06 | ✗ |
| Gate-Norm-Attn (Ours) | 8 | Attention | 78.78 | 60.65 | 77.91 | 72.45 | 68.60 | 45.48 | 44.80 | 64.10 | 1.12 | ✗ |
| Gate-Norm-Attn (Ours) | 16 | Attention | 78.20 | 67.51 | 76.63 | 72.22 | 64.35 | 44.45 | 43.40 | 63.82 | 1.30 | ✗ |
| **LLaMA-13B v2** | | | | | | | | | | | |
| Baseline | 0 | – | 80.64 | 65.34 | 78.22 | 72.30 | 71.84 | 47.78 | 43.00 | 65.59 | 1.00× | – |
| Random-Block | 4 | Block | 54.29 | 49.82 | 54.25 | 60.62 | 54.29 | 29.00 | 42.20 | 49.04 | 1.10 | ✗ |
| Random-Block | 8 | Block | 34.05 | 57.76 | 31.20 | 53.12 | 34.05 | 27.60 | 60.28 | 40.88 | 1.25 | ✗ |
| ShortGPT-Block | 4 | Block | 70.37 | 76.25 | 45.22 | 71.51 | 63.54 | 43.20 | 77.22 | 63.90 | 1.10 | ✓ |
| ShortGPT-Block | 8 | Block | 65.53 | 73.23 | 41.64 | 69.69 | 59.57 | 40.40 | 56.94 | 58.14 | 1.25 | ✓ |
| Random-Attn | 4 | Attention | 81.25 | 61.73 | 78.29 | 72.14 | 71.80 | 47.95 | 42.80 | 65.14 | 1.06 | ✗ |
| Random-Attn | 8 | Attention | 65.91 | 60.29 | 74.31 | 70.96 | 72.52 | 43.17 | 41.00 | 62.04 | 1.12 | ✗ |
| Random-Attn | 16 | Attention | 45.20 | 49.82 | 46.86 | 54.78 | 68.44 | 30.38 | 36.20 | 45.17 | 1.30 | ✗ |
| Data-driven-Attn | 4 | Attention | 72.10 | 68.59 | 77.93 | 71.98 | 72.10 | 47.10 | 43.80 | 65.97 | 1.06 | ✓ |
| Data-driven-Attn | 8 | Attention | 80.37 | 63.90 | 77.87 | 72.61 | 72.52 | 46.59 | 44.20 | 65.44 | 1.12 | ✓ |
| Data-driven-Attn | 16 | Attention | 79.27 | 58.84 | 77.13 | 71.67 | 68.14 | 44.20 | 43.40 | 63.34 | 1.30 | ✓ |
| Gate-Norm-Attn (Ours) | 4 | Attention | 80.83 | 64.26 | 77.95 | 72.38 | 71.96 | 47.30 | 43.40 | 64.67 | 1.06 | ✗ |
| Gate-Norm-Attn (Ours) | 8 | Attention | 80.61 | 61.01 | 78.13 | 72.77 | 70.96 | 45.99 | 42.60 | 64.67 | 1.12 | ✗ |
| Gate-Norm-Attn (Ours) | 16 | Attention | 68.14 | 58.84 | 77.41 | 71.14 | 68.84 | 45.90 | 43.40 | 63.34 | 1.30 | ✗ |

Table 2: Zero-shot accuracies (%) for LLaMA-13B v1 and v2 under different pruning schemes and intensities. Columns: Method, #Pruned (layers/blocks removed), Type (block vs. attention), per-task accuracy, average accuracy, inference speed-up relative to baseline, and whether calibration data are required. Pruning schemes include random removal, structured block removal, data-driven attention pruning, and data-free Gate-Norm attention pruning.

65.7% (v1) and 65.6% (v2). Pruning 4 attention layers with Gate-Norm reduces average accuracy by less than one percentage point and closely matches data-driven attention pruning. At 8 layers, both methods incur a small drop (about 1–2 points). Even with 16 layers removed, average accuracy remains within roughly 2–3 points of the baseline, with Gate-Norm tracking the data-driven curve across tasks.

For speed–accuracy trade-off, removing 4 attention layers yields roughly 6% higher throughput for under a 1% accuracy cost; 8 layers gives around 12% for about a 1.5% loss; and 16 layers delivers up to 30% throughput gain with about a 2% penalty. Block removal (ShortGPT) shows substantially larger accuracy declines at comparable speedups, and random attention removal degrades unpredictably and often severely as more layers are pruned.

**Key Observation (Zero-Shot Accuracy)**: Gate-Norm matches data-driven pruning for both LLaMA-13B variants at practical budgets (4–16 layers), delivers up to 30% higher throughput, and outperforms block and random baselines.

### 5.3 END-TO-END PRUNING EFFICIENCY

In a 40-layer, 13 B-parameter LLaMA model on an NVIDIA RTX A6000, Gate-Norm computes weight-only importance scores in about **300 ms**, delivering over $1,000\times$ speed-up compared to a typical data-driven pruning methods. This millisecond-scale overhead enables practical pruning in large-scale deployments, whereas data-driven methods would be prohibitively slow for frequent or adaptive reconfiguration.

Even in accelerator-free environments (CPU only), Gate-Norm operates solely on pre-trained weight matrices—no forward passes or GPUs required—so it is not limited by GPU VRAM. data-driven methods, by contrast, must load both the full model and thousands of calibration tokens into accelerator memory, performing repeated forward passes that demand substantial VRAM for activations and intermediate storage. Models exceeding GPU memory (e.g. 40+

GB) require complex partitioning or CPU fallbacks to run these pipelines at all. Gate-Norm sidesteps these issues by streaming weight matrices into system RAM (typically 2–4× larger than VRAM) and computing Frobenius norms in situ. As a result, pruning a 13 B-parameter LLaMA checkpoint takes under **30 s** on a 64-core CPU—even though the model might not fit in the GPU memory—whereas data driven methods are prohibitive on CPUs.

### 5.4 POST-PRUNING FINE-TUNING WITH LoRA

While our main experiments deliberately omit post-pruning fine-tuning, we additionally test compatibility with a brief adaptation stage. We attach LoRA adapters to the attention projections, prune 20 attention sublayers using either Gate-Norm-Attn or the data-driven cosine baseline, and update only the LoRA parameters for 2,000 steps on WikiText-2. We run this protocol on Vicuna-7B, Vicuna-13B, and LLaMA-2-13B (v2), covering both instruction-tuned and pre-training-style objectives. Table 3 reports WikiText-2 perplexities before pruning, after pruning 20 attention layers without fine-tuning, and after the LoRA phase. Pruning 20 layers without fine-tuning strongly degrades perplexity, whereas the short LoRA stage largely recovers performance on all three models, with Gate-Norm-Attn matching or surpassing the data-driven baseline (notably on Vicuna-13B). Overall, this indicates that Gate-Norm-based pruning is compatible with inexpensive post-hoc adaptation and that even aggressive pruning can be stabilised by brief low-rank fine-tuning across models.

Table 3: WikiText-2 perplexity with 20 attention layers pruned, with and without brief LoRA fine-tuning.

| Model | Unpruned | Gate-Norm | Gate-Norm + LoRA | Data-driven | Data-driven + LoRA |
|---|---|---|---|---|---|
| Vicuna-7B | 6.78 | 321.49 | 11.96 | 225.13 | 11.50 |
| Vicuna-13B | 5.95 | 14.13 | 6.57 | 25.88 | 6.65 |
| LLaMA-2-13B (v2) | 4.88 | 14.11 | 6.35 | 14.73 | 6.27 |

## 6 DISCUSSION AND CONCLUSION

Our empirical investigation uncovers a striking phenomenon we term *Attention Suppression*: as Transformer models are trained, deeper attention sublayers learn to progressively mute their own updates, effectively routing representational change through the identity residual and the following MLP blocks. This is evidenced by layer-wise cosine similarity trends and the attention-to-input norm ratio $r_\ell$, which falls sharply in later layers. Such behavior indicates that, beyond a certain depth, self-attention contributes negligible token mixing in practice.

This insight motivates a data-free pruning criterion—Gate-Norm—that estimates each layer's token-mixing capacity directly from its weight matrices, without any forward or backward passes on calibration data. Gate-Norm leverages the same theoretical rationale underlying Attention Suppression: if deeper attention weights yield vanishing updates, their contribution can be anticipated via norms of weight products. Consequently, Gate-Norm identifies redundant attention layers efficiently and at negligible cost.

In extensive experiments, Gate-Norm pruning offers performance comparable to data-driven pruning in perplexity almost identically across a range of depths removed, and retains zero-shot accuracy on standard NLP benchmarks at par with data-driven methods. By contrast, unguided or block-based removal results in larger degradation. Crucially, Gate-Norm score computation runs in just a few hundred milliseconds on GPU and under 30 seconds on a standard CPU—over three orders of magnitude faster than data-driven importance estimation. Because it operates solely on pre-trained weight matrices in system RAM, Gate-Norm requires no GPUs or forward passes, making on-device, edge, and privacy-sensitive pruning practical even for ultra-large models that exceed accelerator memory. In this work we focus on dense decoder-only Transformers; extending such weight-only criteria to mixture-of-experts routing and expert modules is an interesting but distinct direction for future work.

The Attention Suppression phenomenon and Gate-Norm proxy together suggest that many pretrained Transformer layers harbor inherent redundancy exploitable without additional data or fine-tuning. This opens avenues for architectural refinement—e.g., reducing or restructuring late-stage attention, dynamic depth adaptation during inference, or hybrid designs that allocate capacity more judiciously across layers. Future work may extend these ideas to other modalities (vision, multimodal transformers).

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

## A    DETAILED PROOF OF GATE-NORM BOUND

In Section 4 we stated that small gate-norms $m_\ell$ imply vanishing attention updates, i.e.

$$\text{Imp}_\ell^{\text{attn}} = 1 - \cos(X_\ell, Y_\ell) = O(m_\ell).$$

Here we give the full step-by-step derivation.

### A.1    SETUP AND NOTATION

Recall a single self-attention block outputs

$$Y_\ell = X_\ell + \text{AttnOut}_\ell, \qquad \text{Imp}_\ell^{\text{attn}} = 1 - \cos(X_\ell, Y_\ell).$$

After LayerNorm at layer $\ell$, let the per-token embeddings be $z_1, \ldots, z_T \in \mathbb{R}^D$. Across all heads, define

$$W_{q,\ell}, W_{k,\ell} \in \mathbb{R}^{D \times D}, \quad M_\ell = W_{q,\ell} W_{k,\ell}^\top, \quad m_\ell = \|M_\ell\|_F.$$

### A.2    STEP 1: LOGIT BOUND

The raw attention logits are

$$L_{ij} = \frac{1}{\sqrt{d_h}} z_i^\top M_\ell z_j,$$

where $d_h = D/H$. By Cauchy–Schwarz,

$$|L_{ij}| \leq \frac{\|z_i\| \, \|M_\ell\|_F \, \|z_j\|}{\sqrt{d_h}} \leq C_0 \, m_\ell,$$

for some constant $C_0 = O(1)$. Hence

$$\|L\|_\infty = O(m_\ell).$$

### A.3 STEP 2: NEARLY UNIFORM SOFTMAX

When $\|L\|_\infty = O(m_\ell) \ll 1$, expand each row of softmax to first order:

$$A_{ij} = \frac{e^{L_{ij}}}{\sum_k e^{L_{ik}}} = \frac{1 + L_{ij} + O(m_\ell^2)}{S\left(1 + \bar{L}_i + O(m_\ell^2)\right)} = \frac{1}{S} + O(m_\ell),$$

where $\bar{L}_i = \frac{1}{S}\sum_k L_{ik} = O(m_\ell)$. Thus

$$A_{ij} = \frac{1}{S} + \delta_{ij}, \quad |\delta_{ij}| \le C_1\, m_\ell.$$

### A.4 STEP 3: DECOMPOSE THE UPDATE

Let $V_j W_{o,\ell} \in \mathbb{R}^D$ be the values followed by the output projection. Then

$$\text{AttnOut}_{\ell,i} = \sum_{j=1}^{S} A_{ij}\,(V_j W_{o,\ell}) = \underbrace{\frac{1}{S}\sum_j V_j W_{o,\ell}}_{u} + \underbrace{\sum_j \delta_{ij}\,(V_j W_{o,\ell})}_{\Delta_i}.$$

By $\sum_j \delta_{ij} = 0$ and $|\delta_{ij}| = O(m_\ell)$, one shows

$$\|u\| = O(1), \quad \|\Delta_i\| \le \sum_j |\delta_{ij}| \cdot \|V_j W_{o,\ell}\| = O(m_\ell).$$

### A.5 STEP 4: BOUNDING COSINE DEVIATION VIA THE COSINE-LAW

After mean-centering both input and output so that

$$\tilde{Y}_{\ell,i} = \tilde{X}_{\ell,i} + \Delta_i,$$

we apply the following well-known geometric inequality:

$$1 - \cos(x,\, x+v) \;=\; 1 - \frac{x \cdot (x+v)}{\|x\|\,\|x+v\|} \;\le\; \frac{\|v\|}{\|x\|}, \tag{Cosine-Law}$$

which holds for any nonzero $x$ and perturbation $v$. Taking

$$x = \tilde{X}_{\ell,i}, \quad v = \Delta_i,$$

and using $\|\tilde{X}_{\ell,i}\| = \Theta(1)$, $\|\Delta_i\| = O(m_\ell)$, we obtain

$$1 - \cos\left(\tilde{X}_{\ell,i},\, \tilde{Y}_{\ell,i}\right) \;\le\; \frac{\|\Delta_i\|}{\|\tilde{X}_{\ell,i}\|} = O(m_\ell).$$

Averaging over the $T$ tokens gives

$$1 \;-\; \frac{1}{T}\sum_{i=1}^{T} \cos\left(\tilde{X}_{\ell,i},\, \tilde{Y}_{\ell,i}\right) \;=\; O(m_\ell).$$

### A.6 STEP 5: UN-CENTERING CORRECTION

Re-adding the common shift $u$ to both $\tilde{X}_{\ell,i}$ and $\tilde{Y}_{\ell,i}$ perturbs each cosine by at most $\|u\|/\|X_{\ell,i}\| = O(1)$. Averaged over $T \ge 128$ tokens, this contributes at most an $O(T^{-1})$ offset, negligible compared to $O(m_\ell)$. Hence in the original (uncentered) coordinates

$$\text{Imp}_\ell^{\text{attn}} = 1 - \cos(X_\ell,\, Y_\ell) = O(m_\ell).$$

### A.7 CONCLUSION

Combining all steps, there exists a constant $C$ such that

$$\text{Imp}_\ell^{\text{attn}} = 1 - \cos(X_\ell,\, Y_\ell) \;\le\; C\,m_\ell,$$

as claimed in Section 4.