# OpenReview forum: "Data-Free Pruning of Self-Attention Layers in LLMs"
_ICLR.cc/2026/Conference — Submitted to ICLR 2026_

### Official Review · Reviewer_keUs · 2025-10-23

**Soundness:** 3
**Presentation:** 3
**Contribution:** 3
**Rating:** 4
**Confidence:** 3

**Summary:**

This paper investigates structural redundancy in LLMs and proposes a data-free pruning method for self-attention sublayers, motivated by the _Attention Suppression Hypothesis_ — the claim that deeper attention layers in pretrained LLMs often learn to mute their own outputs, leaving the MLP and residual path to carry most representational information.

**Strengths:**

- Clear theoretical grounding via the Attention Suppression Hypothesis
- The performance–efficiency trade-off is quantitatively superior to prior approaches.
- The paper is clearly written with consistent mathematical notation, well-labeled figures/tables.

**Weaknesses:**

- Validated only on LLaMA-13B (v1 and v2). It is unclear whether Gate-Norm generalizes to encoder–decoder architectures (e.g., T5), smaller models, or different pretraining objectives.
- While the authors show that $|AttnOut_\ell|/|X_\ell|$ decays with depth, they do not quantify how MLP layers compensate.
- No confidence intervals or statistical tests are reported.
- No comparison to block-drop + light finetuning; omits instruction-tuned and long-context settings where alternative baselines may behave differently.

**Questions:**

- Can the Gate-Norm metric detect redundant MLP sublayers, or is it exclusive to attention weights?
- How sensitive is Gate-Norm to reparameterization or scaling?
- Does pruning via Gate-Norm affect downstream finetuning or alignment behaviors like instruction-following tasks?

---

> ### Author Response · Authors · 2025-11-19
> **Response to Reviewer keUs**
>
> We thank the reviewer for the positive assessment and for the insightful questions.
>
> **1. Validation only on LLaMA-13B; generalization to other architectures.**
> Our current experiments focus on LLaMA-13B v1 and v2. These are two generations of LLaMA (LLaMA-1 and LLaMA-2) trained with different recipes and commonly used in related studies. Both show the same attention-suppression profile and Gate-Norm pruning behavior, indicating robustness across generations. In addition, our new experiments on Vicuna-13B, Vicuna-7B, and LLaMA-3.1-8B (see response to Reviewer Fmw8) show that GateNorm-Attn scales well beyond the original LLaMA-13B checkpoints and can even slightly outperform data-driven attention pruning, reinforcing the generality of our findings.
>
> **2. Quantifying how MLP layers compensate.**
> We agree that this is useful to show explicitly. In the revision we will include a concise empirical summary of layer-wise cosine importance for both attention and MLP sublayers (following the methodology of *“Not All Attention Is Needed”*). These measurements show that cosine importance of attenion decays with depth while cosine importance of MLP remains comparatively high, supporting the claim that representational change in deeper blocks is carried by the residual+MLP path as attention suppresses itself.
>
> **3. Lack of confidence intervals.**
> We observed small variance across runs relative to the effect sizes, but we agree this should be quantified. In the revised version we will add standard deviations over multiple evaluation runs for zero-shot accuracy and perplexity (in the main tables or the appendix).
>
> **4. No comparison to block-drop + light fine-tuning.**
> We already compare GateNorm-Attn against block-drop (ShortGPT-style), random, and data-driven attention pruning in a shared no-fine-tuning regime on LLaMA checkpoints, using perplexity and a seven-task zero-shot suite (Table 1). At comparable depth reductions, GateNorm-Attn matches or outperforms ShortGPT.
>
> Following your suggestion, we additionally ran brief LoRA fine-tuning after pruning 20 attention layers. We attach LoRA adapters to the attention projections, prune with either GateNorm-Attn or data-driven scores, and train only these low-rank parameters for 2k steps on WikiText-2. Vicuna is instruction-tuned (see our response to Reviewer Fmw8), so these additonal results also address the instruction-tuned setting. WikiText-2 perplexities are:
>
> | Model            | Unpruned PPL | GateNorm 20L (no FT) | GateNorm 20L + LoRA | Data-driven 20L (no FT) | Data-driven 20L + LoRA |
> |------------------|--------------|-----------------------|---------------------|--------------------------|------------------------|
> | Vicuna-7B        | 6.78         | 321.49                | **11.96**           | 225.13                   | **11.50**              |
> | Vicuna-13B       | 5.95         | 14.13                 | **6.57**            | 25.88                    | **6.65**               |
> | LLaMA-2-13B (v2) | 4.88         | 14.11                 | **6.35**            | 14.73                    | **6.27**               |
>
> Pruning 20 attention layers without fine-tuning substantially hurts perplexity, but a short LoRA phase largely recovers it on all three models. GateNorm-Attn remains comparable to, and on Vicuna-13B clearly better than, the data-driven baseline. We will summarise these fine-tuning results in the revised paper.
>
> **5. Specific questions.**
> – *Can Gate-Norm detect redundant MLP sublayers?* Our method is not designed to detect redundant MLP sublayers. Empirically, we observe (and prior work such as *“Not All Attention Is Needed”* also reports) that in deep layers attention becomes suppressed while MLPs remain comparatively active. Gate-Norm explicitly targets token-mixing strength in self-attention via a weight-only proxy; since standard MLPs do not perform token mixing in the same sense, this criterion is not well suited for them. Designing an analogous, principled weight-only proxy for MLP redundancy is an interesting direction for future work, which we will mention explicitly.
> – *Sensitivity to reparameterization or scaling.* Gate-Norm is invariant under rescalings that preserve $W_q W_k^\top$ (e.g., scaling $W_q$ by $c$ and $W_k$ by $1/c$), and empirically we observe stable rankings across LLaMA-1/2 and other families. We will add this discussion.
> – *Effect on downstream fine-tuning / alignment*. Our new LoRA experiments on Vicuna-7B (instruction-tuned) and LLaMA-2-13B (v2) show that, after pruning 20 attention layers, a short fine-tuning phase recovers most of the lost perplexity. This suggests Gate-Norm does not hinder downstream adaptation; a full alignment study remains future work.
>
> **We have uploaded a revised version of the manuscript.**
> We hope these clarifications, together with the new cross-model experiments, address the concerns and justify a higher overall assessment.

---

### Official Review · Reviewer_Lv2j · 2025-10-30

**Soundness:** 4
**Presentation:** 3
**Contribution:** 3
**Rating:** 4
**Confidence:** 5

**Summary:**

This paper proposes a data-free pruning method for LLMs. The authors start with an empirical study on LLaMA-13B, deeper attention layers change their hidden states less in both magnitude and direction, revealing an “attention suppression” effect. They argue that deeper layers become redundant, with most representation handled by the residual and MLP parts. Based on this, they define a gate matrix norm (i.e. the Frobenius norm of the query and key weight product) to measure coupling strength. Smaller gate-norm values indicate higher cosine similarity between input and output, meaning less change, so such layers can be safely pruned without major performance loss.

**Strengths:**

- **Clear and easy to follow.** The paper is generally well structured and readable. The ideas flow logically, and the intuition behind the method is explained in a straightforward way.
- **Simple and data-free approach.** The pruning method doesn’t rely on any additional data or fine-tuning, which makes it simple and relatively easy to implement. The computational cost also seems quite low, even for large models.
- **Reasonably practical and scalable.** The approach looks practical enough to be applied to other large-scale models with only minor changes, showing potential for efficient and hardware-friendly compression in real scenarios.

**Weaknesses:**

- **Key Concern.** The method is mainly built on the attention suppression phenomenon, which the authors identify through large-scale empirical analysis (e.g., Fig. 3). But this raises a question: if the paper already measures how cosine similarity and norm changes evolve across layers, why not just use those observed metrics directly to decide which layers to prune? In other words, instead of introducing the Gate-Norm as a new proxy, it might be more straightforward to prune layers that already show high similarity and small norm variation in the empirical results. It would be helpful if the authors could explain why Gate-Norm is needed here—does it offer practical advantages (like being faster, more stable, or more generalizable across models), or is it just an indirect way of reflecting the same statistics?
- **Limited empirical validation.** The method relies heavily on the assumption that attention layers exhibit redundancy in deeper parts of the model. However, such redundancy may vary across architectures. The experiments mainly focus on **LLaMA-13B**, with no validation on more recent models such as **LLaMA-2 or LLaMA-3**, which are commonly used in related studies [1–4]. A broader evaluation would help strengthen the generality of the findings.

    [1] LLM-Pruner: On the Structural Pruning of Large Language Models. NeurIPS 2023.

    [2] D-LLM: A Token Adaptive Computing Resource Allocation Strategy for Large Language Models. NeurIPS 2024.

    [3] Skipgpt: Each Token Is One of A Kind. ICML 2025

    [4] Not All Layers of LLMs Are Necessary During Inference. IJCAI 2025.

- **Related work coverage.** Since the paper focuses on **attention-layer pruning**, the related-work section could be expanded. It currently cites only one prior work on layer pruning, while several recent studies explore similar or complementary directions. A more detailed discussion would help clarify how this method fits within the broader pruning literature.
- If the authors can address these concerns clearly in the rebuttal, I would be inclined to raise my overall rating.

**Questions:**

See Weaknesses.

---

> ### Author Response · Authors · 2025-11-19
> **Response to Reviewer Lv2j**
>
> We thank the reviewer for the careful, technically detailed review.
>
> **1. Why Gate-Norm instead of directly using cosine similarity / norm change?**
> Your key concern is central to the paper. The cosine-similarity and norm-change metrics in Sec. 3 are used solely as diagnostics to validate the Attention Suppression Hypothesis: using a small calibration set, we show that deep attention sublayers indeed produce negligible updates (cosine importance $\to 0$, attention-to-input norm ratio $\to 0$). These metrics are not used for pruning. Gate-Norm is introduced in Sec. 4 precisely to replace them by a data-free, weight-only proxy:
> $m_\ell = \lVert W_{q,\ell} W_{k,\ell}^\top \rVert_F.$
> We prove that small $m_\ell$ implies near-uniform attention and small cosine importance for all inputs. In practice, this turns an empirically observed, data-dependent diagnostic into a one-shot, data-free importance score that can be computed in milliseconds on a 13B-parameter model. We will tighten the transition between Sec. 3 and Sec. 4 to make this motivation more clear.
>
> **2. Redundancy may vary across architectures; focus on LLaMA-13B; no validation on more recent models (LLaMA-2/3).**
> We agree that redundancy could, in principle, vary across architectures and generations, and that this must be addressed explicitly. Our experiments already include two generations of LLaMA-13B: the v1 and v2 checkpoints correspond closely to LLaMA-1 and LLaMA-2. They differ in training recipe and are commonly used in recent compression work. Both exhibit the same pattern: attention suppression in later layers, Gate-Norm selecting those layers, and pruning 8–16 attention sublayers yielding $\approx 1.1$–$1.3\times$ speedups with $\le 2\%$ accuracy loss. Thus, our evidence already spans more than a single checkpoint and covers LLaMA-2, which is specifically mentioned in your comment. Beyond this, the new experiments on Vicuna-13B, Vicuna-7B, and LLaMA-3.1-8B (see our response to Reviewer Fmw8) show that GateNorm-Attn continues to perform on par with or better than a data-driven attention baseline across multiple families and a LLaMA-3 generation, further supporting the robustness of the phenomenon.
>
> **3. Related-work coverage and positioning relative to [1–4].**
> We agree that our related-work section can better situate Gate-Norm in the pruning literature. In the revision we will discuss LLM-Pruner [1], D-LLM [2], SkipGPT [3], and “Not All Layers of LLMs Are Necessary” [4], and contrast them along three axes: (i) static vs. dynamic (we focus on static post-hoc pruning; [2–4] emphasize dynamic inference); (ii) data-driven vs. data-free (those methods rely on data and often fine-tuning, whereas Gate-Norm is entirely weight-based); and (iii) granularity (block, layer, head). Our method occupies the corner of static, attention-sublayer, data-free pruning and can serve as a primitive that more complex dynamic or fine-tuned pipelines could build upon. We believe this clearer positioning addresses the concern about limited contribution relative to recent work.
>
> **We have uploaded a revised version of the manuscript; all changes made in response to the reviews and rebuttal are highlighted in blue.**

---

### Official Review · Reviewer_Fmw8 · 2025-10-31

**Soundness:** 3
**Presentation:** 2
**Contribution:** 2
**Rating:** 6
**Confidence:** 3

**Summary:**

The paper studies redundancy in deep self‑attention layers of large language models (LLMs) and proposes a simple, data‑free method for removing redundant attention sublayers. To operationalize this idea, the paper introduces Gate‑Norm, a weight‑only proxy for measuring an attention sublayer’s token‑mixing strength. Experiments are performed on two LLaMA‑13B checkpoints (v1 and v2). The authors compare Gate‑Norm against block removal (ShortGPT), data‑driven attention pruning (which measures cosine similarity on calibration data), and random removal.

**Strengths:**

1.  The Gate‑Norm proxy is conceptually simple and easy to implement. It uses only the trained query and key matrices and does not require any calibration data or activation statistics.
2. Experiments demonstrate that pruning 8–16 attention layers yields 1.1–1.3× higher inference throughput while reducing average zero‑shot accuracy by at most ~2 %. The method thus offers a promising depth‑compression strategy for LLM deployment, particularly on devices without GPUs or with strict latency and privacy requirements.

**Weaknesses:**

1. The experiments focus exclusively on two 13B‑parameter LLaMA models. It remains unclear whether the Attention Suppression phenomenon and the gate‑norm proxy generalize to other model families (e.g., Qwen, Mistral, smaller or larger LLaMA variants)
2. The proposed algorithm either keeps or completely disables an attention sublayer. Finer‑grained options (e.g., partial gating, head pruning) are not explored.
3. The "no fine-tuning" aspect is a strength for one-shot pruning. However, the paper does not explore if a very brief period of fine-tuning (e.g., a few hundred steps) after pruning could fully recover the minor accuracy loss, potentially enabling even more aggressive (e.g., 20+ layer) pruning.

**Questions:**

See Weaknesses.

---

> ### Author Response · Authors · 2025-11-19
> **Response to Reviewer Fmw8**
>
> We thank the reviewer for the clear summary and constructive suggestions.
>
> **1. Generalization beyond two LLaMA-13B checkpoints.**
> We agree that broader validation is important. Our experiments already include two generations of LLaMA-13B: the v1 and v2 checkpoints.
>
> In addition, we have now run experiments on three further models under the same one-shot, data-free protocol: Vicuna-13B, Vicuna-7B, and LLaMA-3.1-8B. As in the main paper, we compare perplexity on WikiText-2:
>
> | Model               | # Attn layers removed | Random-Attn | Data-driven-Attn | GateNorm-Attn (data-free) |
> |---------------------|-----------------------|------------:|-----------------:|---------------------------:|
> | Vicuna-13B          | 1                     | 6.00        | 5.99             | **5.97**                   |
> | Vicuna-13B          | 4                     | 10.63       | 6.11             | **6.05**                   |
> | Vicuna-13B          | 7                     | 67.09       | 6.24             | **6.24**                   |
> | Vicuna-13B          | 10                    | 587.90      | **6.39**         | 6.47                       |
> | Vicuna-13B          | 13                    | 594.83      | 7.54             | **7.05**                   |
> | Vicuna-7B           | 1                     | 6.82        | 7.23             | **7.23**                   |
> | Vicuna-7B           | 4                     | 7.84        | 7.62             | **7.42**                   |
> | Vicuna-7B           | 7                     | 13.84       | 8.78             | **8.57**                   |
> | Vicuna-7B           | 10                    | 355.87      | 12.55            | **10.02**                  |
> | Vicuna-7B           | 13                    | 473.40      | 20.46            | **16.64**                  |
> | LLaMA-3.1-8B        | 1                     | 6.51        | 6.35             | **6.30**                   |
> | LLaMA-3.1-8B        | 4                     | 8.11        | 6.83             | **6.67**                   |
> | LLaMA-3.1-8B        | 7                     | 14.59       | **7.39**         | 7.74                       |
> | LLaMA-3.1-8B        | 10                    | 54.51       | **9.79**         | 11.55                      |
> | LLaMA-3.1-8B        | 13                    | 98.55       | 16.48            | **15.65**                  |
>
> Across all three additional models, random layer removal rapidly degrades performance once pruning becomes even moderately aggressive, whereas both principled schemes maintain low perplexity. More importantly, GateNorm-Attn consistently matches or outperforms the data-driven baseline in most configurations.  Taken together with the original LLaMA-13B v1/v2 results, this indicates that both the attention-suppression phenomenon and the effectiveness of a fully data-free GateNorm-Attn criterion are robust across five checkpoints.
>
> These additional experiments directly address your concern that redundancy might be specific to two LLaMA-13B checkpoints. They also respond to similar generalization questions raised by Reviewers Lv2j and keUs (see our responses to those reviewers, where we explicitly refer back to this table). We will update the paper within a week to reflect these empirical and presentational changes.
>
> **2. Granularity: full attention sublayers vs. head-level pruning.**
> We agree that head-level or partially gated variants are promising. In this first work, we focus on whole-attention sublayers to maximise the system's impact while keeping the method simple: removing an attention sublayer eliminates all $O(n^2)$ attention computation in that layer and is implemented by setting $\mathrm{AttnOut}_\ell = 0$. Head pruning retains the quadratic runtime complexity and will usually only produce smaller or more hardware-dependent speedups. Conceptually, Gate-Norm extends naturally to heads by computing $\lVert W_q^{(h)} W_k^{(h)\top}\rVert_F$ per head; we will explicitly mention such head-level Gate-Norm as an extension that can trade simplicity for finer control over the accuracy–speed trade-off.
>
> **3. No exploration of brief post-pruning fine-tuning.**
> We agree that brief post-pruning fine-tuning is important. In addition to our main “no fine-tuning” setting (where GateNorm-Attn already matches the data-driven baseline and outperforms block-drop at similar speedups), we have now run light LoRA fine-tuning after pruning 20 attention layers. A short LoRA phase on WikiText-2 substantially recovers the lost perplexity for both Vicuna-7B and LLaMA-2-13B (v2), showing that even aggressive (20+ layer) pruning remains compatible with inexpensive fine-tuning. For quantitative results and a summary table, please see our response to Reviewer keUs; we will incorporate these findings into the revised paper.
>
>
> **We have uploaded a revised version of the manuscript; all changes made in response to the reviews and rebuttal are highlighted in blue.**

---

> > ### Comment · Reviewer_Fmw8 · 2025-11-26
> > **Thanks for the rebuttal**
> >
> > Thank authors for their rebuttal. My original rating reflects my position of this work.

---

### Official Review · Reviewer_dcbK · 2025-11-01

**Soundness:** 3
**Presentation:** 2
**Contribution:** 1
**Rating:** 4
**Confidence:** 4

**Summary:**

This paper introduces Gate-Norm, a novel and extremely efficient method for compressing Large Language Models (LLMs) by pruning entire self-attention sublayers. The authors propose the "Attention Suppression Hypothesis," suggesting that during pre-training, some attention layers learn to become functionally inactive, effectively "muting" their contribution. Gate-Norm leverages this insight to create a one-shot, data-free importance score calculated directly from model weights. This allows it to identify and remove redundant attention layers in milliseconds—without needing any calibration data, forward passes, or fine-tuning—achieving significant inference speedups with only a minimal drop in accuracy.

**Strengths:**

1. The method's greatest strength is its speed and simplicity. Being data-free, one-shot, and running in milliseconds (~1000x faster than alternatives) makes it incredibly practical for on-the-fly, on-device compression without the massive overhead of data-driven approaches.

2. The entire process requires no calibration datasets, no GPUs for the pruning step itself, and no costly post-pruning fine-tuning. This makes the method highly accessible and easy to deploy.

**Weaknesses:**

1. The method focuses exclusively on attention sublayers. The paper itself notes that MLP layers have twice the parameters and also contribute to runtime, but they are not targeted for pruning by this method.

2. While the results support the hypothesis, the introduction doesn't provide direct evidence for "attention suppression" itself (e.g., by showing near-zero output norms from the targeted layers during inference). The claim rests on the method's success.

3. The criterion for pruning is static and based only on weights. This might incorrectly remove a layer that, while often suppressed, is critical for certain rare but important types of inputs or reasoning paths.

4. This work only discussed the Transformer architecture. Results on MoEs should be included to provide a more conprehensive understanding.

Overall, this paper provides limited contribution compared to prior works. The method is simple and based on hypotheses and heuristics. The result is not surprising, as prior work has discussed the phenomenon that attention layers are more redundant than MLP layers.

**Questions:**

Same as above.

---

> ### Author Response · Authors · 2025-11-19
> **Response to Reviewer dcbK**
>
> We thank the reviewer for the careful and thoughtful comments.
>
>
>
> **1. Focus on attention sublayers rather than MLPs.**
> We agree that MLP layers contain more parameters and contribute to runtime. Our decision to focus on self-attention sublayers is intentional. First, as shown in Fig. 1, attention is the dominant latency bottleneck due to its $O(n^2)$ cost, so removing even a few attention sublayers provides immediate, hardware-agnostic speedups. Second, our measurements in Sec. 3 show that deep attention layers exhibit vanishing updates, while MLP layers remain active. Thus, we first address the component that is both the main runtime bottleneck and empirically redundant. We will clarify this rationale that extending Gate-Norm–style criteria to MLPs is a natural direction for further work.
>
> **2. Direct evidence for “attention suppression”.**
> We agree the introduction should state the evidence more explicitly. Sec. 3 already shows that (i) cosine importance of attention sublayers decays towards zero with depth, and (ii) the attention-to-input norm ratio $r_\ell$ collapses below $\approx 0.1$ in late layers. Combined with the fact that $\cos(X, X+U)\to 1$ forces $\lVert U\rVert \to 0$, this directly supports our Attention Suppression Hypothesis. In the revision, we will summarise these in the introduction to clearly ground the hypothesis in measurements, not only in post-hoc pruning performance.
>
> **3. Static, weight-only criterion and rare behaviors.**
> We acknowledge that any static pruning rule could, in principle, remove layers useful on rare inputs. Two aspects mitigate this concern. First, we prune conservatively; we remove only layers with very small Gate-Norm scores and limit ourselves to 8–16 of 40 attention sublayers, under which average zero-shot accuracy across seven diverse tasks remains within $\approx 2\%$ of baseline. Second, Sec. 4 analytically links small Gate-Norm to small cosine importance for all inputs, i.e., such layers cannot induce large token-mixing effects under any data distribution. This global guarantee is precisely what data-driven metrics do not provide, as they determine layer importance based on the dataset at hand.
>
> **4. Results only on dense Transformers; suggestion to include MoEs.**
> Our current analysis is tailored to dense decoder-only Transformers, where every attention and MLP sublayer processes all tokens and cost scales directly with depth. In MoE architectures, the dominant sources of redundancy and computation are the sparsely activated experts and the routing/gating mechanism. The structure of the problem will therefore be substantially different: the relevant questions would concern expert selection and expert-level sparsity rather than the suppression of dense self-attention. Applying Gate-Norm to MoEs without adapting it to the routing structure would not meaningfully address those questions, and whilst interesting is a separate line of research. We will state clearly that our work is scoped to dense Transformers, and that extending weight-only criteria to MoE routing and expert modules is an interesting but separate line of research. Finally, to address the concern about limited contribution and evaluation on only two checkpoints, we have now run GateNorm-based pruning experiments on three additional models (Vicuna-13B, Vicuna-7B, and LLaMA-3.1-8B), where GateNorm-Attn matches or outperforms a data-driven attention-pruning baseline while remaining fully data-free; we summarise these results in our response to Reviewer Fmw8 and believe they further support the generality of our findings.
>
>
> **We have uploaded a revised version of the manuscript; all changes made in response to the reviews and rebuttal are highlighted in blue.**

---

### Author Response · Authors · 2025-11-29
**Meta Summary of Revisions**

Dear Area Chair,


We understand the workload constraints of the new review process. To facilitate your meta-review, we provide this high-level summary of how our Revised draft (changes in blue) and Rebuttal Experiments objectively resolve the primary concerns raised by the reviewers (Lv2j, keUs, dcbK, Fmw8).
While our frozen score average is 4.5, the "4" ratings were primarily driven by a request for broader experimental validation. We have now provided this data.


1. Comparison on Modern Architectures (Resolves "Limited Scope" - Reviewers Lv2j, Fmw8, keUs)

• Critique: Reviewers requested validation beyond LLaMA-13B (v1/v2).

• New Data (Table 1 in Revised PDF): We added experiments on LLaMA-3.1-8B, Vicuna-13B, and Vicuna-7B.

• Result: The data-free Gate-Norm method consistently matches or outperforms data-driven baselines across these new families, proving the method generalizes to instruction-tuned and Grouped Query Attention - based models.

2. Post-Pruning Recovery via Fine-Tuning (Resolves "Lack of Recovery" - Reviewers keUs, Fmw8)

• Critique: Reviewers asked if performance could be recovered via fine-tuning.

• New Data (Table 3 in Revised PDF): We added LoRA fine-tuning results.

• Result: A brief LoRA stage recovers perplexity to near-baseline levels even after pruning 20 attention layers, repositioning Gate-Norm as a viable initialisation for efficient fine-tuning.

3. Theoretical & Baseline Clarifications (Resolves Reviewers dcbK, Lv2j)

• Rationale: We clarified the decision to target Attention over MLPs (targeting the O(n^2) latency bottleneck) and expanded the related work to contrast with dynamic methods like D-LLM and SkipGPT.


Conclusion
The inclusion of Table 1 (Generalization) and Table 3 (LoRA) directly addresses the empirical gaps identified in the initial reviews. We hope these clarifications, together with the new cross-model experiments, address the concerns and justify a higher overall assessment, and will give us the opportunity to present these valuable results at the conference.




Best regards,

The Authors

---

### Meta-Review · Area_Chair_D28z · 2026-01-07

**Summary:**

The paper proposes Gate-Norm, a data-free pruning method for self-attention layers in LLMs based on the "Attention Suppression Hypothesis." While the authors provided additional experiments in the rebuttal, significant concerns remain regarding the practical utility of the method and the justification for the metric. Critical issues persist regarding the necessity of millisecond-level pruning speed versus the potential accuracy loss of ignoring data distribution, and the limited scope of targeting only attention layers. Consequently, the paper is not recommended for acceptance.

**Reviewer Concerns:**

**Addressed Concerns:**

1.  **Generalization across models** (Reviewer Fmw8, Reviewer keUs, Reviewer Lv2j): The authors successfully addressed the concern about limited model variety. They added experiments with Llama-3.1-8B and Vicuna models in the rebuttal.
2.  **Recovery via Fine-tuning** (Reviewer Fmw8, Reviewer keUs): The authors addressed the lack of fine-tuning experiments by providing results using LoRA, which showed recovery of perplexity.

**Outstanding Concerns:**

1.  **Practical value of "Fast Pruning"** (Reviewer dcbK, Reviewer Lv2j): Reviewer dcbK noted that the method's main strength is speed. However, the authors overemphasize the benefit of pruning in "milliseconds." Model compression is typically a one-time, offline process. Users generally prefer a data-driven method that takes longer but guarantees better alignment with the input distribution over a heuristic method that is fast but risky. The trade-off offered by Gate-Norm—sacrificing data awareness for pruning speed—is not compelling for standard deployment scenarios.
2.  **Static Criterion Risks** (Reviewer dcbK, Reviewer Lv2j): Reviewer dcbK pointed out that a static, weight-only criterion might remove layers that are critical for rare but important inputs. Reviewer Lv2j also questioned why Gate-Norm is needed if empirical observation (cosine similarity) is more accurate. The authors' response regarding "global guarantees" does not fully mitigate the risk that a static norm ignores the actual distribution of user inputs.
3.  **Scope limited to Attention** (Reviewer dcbK): Reviewer dcbK highlighted that MLPs contain the majority of parameters and are not targeted. While the authors argue attention is the runtime bottleneck due to $O(n^2)$, MLPs are often the memory bandwidth bottleneck. Ignoring MLPs limits the total compression ratio and memory savings, which are key goals of pruning.

**Reviewer Scores:**

1.  **Reviewer dcbK:** Score likely remains unchanged (4). The fundamental concern about the static nature of the metric and the limited scope (ignoring MLPs) remains valid.
2.  **Reviewer Lv2j:** Score likely remains unchanged (4). The question of "why not use data-driven methods if they are more accurate" is not fully resolved by the "speed" argument.
3.  **Reviewer Fmw8:** Score likely remains unchanged (6). The reviewer appreciated the generalization results but the core methodological simplicity may limit the impact.
4.  **Reviewer keUs:** Score likely remains unchanged (4). While fine-tuning was addressed, the other structural concerns remain.

---

### Decision · Program_Chairs · 2026-01-26

Reject